# Performance- and Resistance-Related Early Responses of Colombian Elite Rubber Tree Genotypes under Low Pressure of South American Leaf Blight: Implications for Disease Management in the Amazon

**DOI:** 10.3390/plants12203627

**Published:** 2023-10-20

**Authors:** Lyda Constanza Galindo-Rodríguez, Armando Sterling, Herminton Muñoz-Ramirez, Jesica Andrea Fonseca-Restrepo

**Affiliations:** 1Doctoral Program in Natural Sciences and Sustainable Development, Faculty of Agricultural Sciences, Universidad de la Amazonia, Florencia 180001, Colombia; 2Laboratory of Phytopathology, Amazonian Scientific Research Institute Sinchi, Faculty of Basic Sciences, Universidad de la Amazonia, Florencia 180001, Colombia; jesicafonsecar38@gmail.com; 3Mycology and Phytoprotection Laboratory, Faculty of Basic Sciences, Universidad de la Amazonia, Florencia 180001, Colombia; hermintonr@gmail.com

**Keywords:** *Hevea brasiliensis*, Pseudocercospora ulei, South American leaf blight, genetic resistance, early performance, clone selection

## Abstract

The cultivation of *Hevea brasiliensis*, the primary commercial source of natural rubber, is strongly impacted by South American leaf blight (SALB) disease, caused by the fungus *Pseudocercospora ulei*. Various management strategies have been implemented, including the selection of resistant genotypes and the identification of escape zones. This study evaluated the growth, early yield, and resistance to SALB of nine Colombian elite genotypes from the ECC-100 series and IAN 873 clone (control) in a large-scale clone trial in an area with low SALB pressure in the Colombian Amazon during 2017–2020. Favorable early performance was evident, although there was a significant increase in the severity and sporulation of *P. ulei* over time, especially in the ECC 35, ECC 60, and IAN 873 genotypes. However, these scores indicate low susceptibility. Genotypes with higher resistance to SALB demonstrated greater growth and early yield compared to more highly susceptible genotypes. The ECC 64, ECC 73, ECC 90, ECC 25, and ECC 29 genotypes were more desirable in low SALB pressure zones due to their higher resistance and early performance. It is important to highlight that this research contributes to the selection of new SALB-resistant Colombian genotypes of *H. brasiliensis*. However, it is also necessary to evaluate the productivity of these selections in the mature stage and long-term resistance to SALB before recommending and promoting their commercial adoption in the Colombian Amazon.

## 1. Introduction

*Hevea brasilienses* (Willd. ex A.Juss.) Müll.Arg. is the primary commercial source of natural rubber (NR) and supports the livelihoods of approximately 20 million people worldwide [1]. This resource is essential as a raw material in the manufacturing of over 50,000 products across various industrial sectors, including healthcare, construction, and automotive. In the latter sector, its usage is particularly prominent in the production of high-strength tires and rubber products [2,3].

Between 1961 and 2020, there was a remarkable increase in the global average yield of rubber cultivation, rising from 0.5 to 1.2 tons per hectare, and annual production surged from 2.1 to 14.8 million tons [4]. This increase as achieved through the implementation of advanced agronomic practices, the careful selection of highly productive and disease-resistant genotypes, the identification of suitable cultivation areas, and the expansion of agricultural land dedicated to rubber tree cultivation [5,6]. The scarcity of other commercial sources of NR offering comparable quality and production capacity has justified research into various aspects of *H. brasiliensis* cultivation [7,8]. In contrast, rubber tree cultivation faces various phytosanitary limitations that, if not addressed in a timely manner, can lead to significant economic losses. In South America, the primary disease affecting *H. brasiliensis* is South American leaf blight (SALB), caused by the fungus *Pseudocercospora ulei* (Henn.) [9].

*P. ulei* is an exclusive hemibiotrophic fungus of the *Hevea* genus [10]. This fungus has three developmental stages in its life cycle. The first stage corresponds to the conidial phase (i.e., asexual phase), which is responsible for the infectious stage of the disease. The lesions caused during the fungus’s development primarily appear on the leaves, stems, petioles, and young fruits. On young leaves, there are spots on the underside that spread across the entire leaf surface, varying in coffee-gray color, shape, and size, with a diameter ranging from 8 to 10 mm [11,12,13,14]. The second stage, corresponding to the onset of the sexual phase (i.e., spermogonial phase) in leaves at the beginning of physiological maturity, is not relevant for the pathogen’s dissemination [15,16]. Finally, the third stage (i.e., sexual phase) occurs in leaves at physiological maturity, with structures of resistance known as stromata [17,18]. These stromata are brown, globular, usually abaxial, carbonaceous, and superficially clustered, and sometimes form rings around the perforations of the leaves, with a size of between 200 and 400 mm [18].

This disease results in repeated defoliations and reduced latex production, and, in severe cases, it can lead to the death of the plant [19], thereby limiting production in the Amazon region [20].

Chemical management of SALB is possible; however, due to the height of the trees (20–30 m) and the required treatment frequency, it is economically unfeasible and raises significant ecological and public health concerns [21]. For the management of SALB, the primary approaches implemented have been the selection of fungus-resistant clones [22] and the identification of escape zones (areas where rubber grows with lower fungal pressure) [1]. Escape zones are areas where the pressure of the *P. ulei* fungus is lower, allowing for the normal development and production of *H. brasiliensis* [23]. These areas have a well-defined dry season that can last from one to five months, a period during which the fungus’s life cycle is interrupted, as the pathogen cannot find favorable conditions for germination, infect receptive tissue, promote the development of the infection process, and subsequently cause damage. This enables the trees to escape the disease, especially during the natural defoliation–refoliation period [24,25]. In addition to the dry season, other climatic variables are considered, such as annual average temperature, evapotranspiration, water deficit, relative humidity, altitude, and soil suitability in the area [9,23,25,26]. This strategy has been implemented in Brazil [27,28], Ecuador [1,25], and Colombia [9,23,26], where the production of natural rubber has been severely impacted by SALB incidence [29,30,31].

According to Food and Agriculture Organization (FAO) data [4], the yield of rubber tree plantations in Colombia has reached 1.6 tons per hectare, surpassing the global average of 1.2 tons per hectare but falling below the averages of plantations in Mexico, Guatemala, and India, where average yields have reached 4.0, 2.8, and 2.1 tons per hectare, respectively. In the Amazon, despite the introduction of genotypes with good resistance and yield characteristics, the cultivation continues to be heavily impacted by the disease, which diminishes productivity [32,33]. Therefore, it is necessary to select new high-yield clones that are more resistant to SALB.

With the establishment of the rubber tree clone trials network in Latin America [1], clones/genotypes that exhibit partial or complete resistance to SALB have been identified [24]. Previous research conducted in the Colombian Amazon has identified superior genotypes in terms of performance, quality, and adaptability to climatic variations [31,34]. For the Elite Caquetá Colombia (ECC) series, Sterling et al. [30] observed favorable early growth in some genotypes in non-SALB escape zones. However, the genotypes with high resistance showed lower growth rates compared to the more susceptible genotypes. Therefore, further research is necessary for the final selection and commercial adoption of resistant genotypes with desirable productivity traits under high and low SALB pressure.

In Colombia, there has been an effort to expand the genetic base of the rubber tree by assessing the growth, physiology, latex production, and response to SALB of introduced clones and genotypes from the ECC-100 series in small-scale and large-scale clone trials under non-SALB escape conditions [30]. However, in escape zones or areas with low disease pressure, it is expected that the environmental response of these genotypes in agronomic and phytosanitary terms will be more favorable. Therefore, the objective of this study was to evaluate the growth, early yield, and SALB resistance of nine Colombian elite genotypes from the ECC-100 series and IAN 873 (control) under low disease pressure in the Colombian Amazon region.

## 2. Results

### 2.1. Climate

Between July 2017 and June 2020, the study area recorded an average monthly precipitation of 209.5 mm and an annual average precipitation of 2514.6 mm. The average annual temperature was 25.1 °C. The maximum temperature ranged from 28.5 °C in June and July 2019 to 34.1 °C in January 2020, whereas the minimum temperature ranged from 21.84 °C in December 2019 to 23.6 °C in February 2020. The average thermal amplitude was 8.7 °C, with an average maximum temperature of 31.6 °C and an average minimum temperature of 22.7 °C. The lowest relative humidity was recorded in January 2020 with a value of 69.3%, whereas the highest occurred in June 2019 with a value of 88.8%, and the annual average was of 81.5%. The dew point ranged from 19.4 °C in January 2020 to 22.9 °C in May 2018 (Figure 1).

### 2.2. Effects of the Fixed Factors

The Akaike (AIC) criterion was minimized in the unstructured with heterogeneous variances (UNH) model for all measured variables except for the type of reaction (TR), for which the first-order autoregressive with heterogeneous variances (ARH1) model performed better in terms of the AIC criterion. For the Bayesian criterion (BIC), the best model was UNH for the circumference of the trunk in the pre-tapping phase (CTpt), the severity of the attack on young leaves (AT1), and TR variables. The ARH1 model was superior for the BIC criterion in terms of the severity of the attack on mature leaves (AT2), and compound symmetry with heterogeneous variances (CSH) model was the best for the BIC in terms of stroma density (ST) (Table 1). Based on the above, the unstructured model with heterogeneous variances (UNH) was the best model. Consequently, significant differences were observed between the genotypes, between the years, and in the genotype × year interaction (*p* < 0.05) (Table 2).

### 2.3. Growth and Early Yield

The annual tree growth data are shown in Table 3. Significant differences were found between the 10 genotypes (*p* < 0.05). Higher mean values of CTpt were observed in the ECC 64, ECC 25, and ECC 83 genotypes, where the ECC 64 genotype had the highest average CTpt over the three years of evaluation (*p* < 0.05). The ECC 25 genotype showed higher means than the other genotypes in years 2 and 3 (*p* < 0.05), where the ECC 64 and ECC 25 genotypes had the greatest vigor (Table 3).

The CTo ranged from 25.00 cm (ECC 35) to 27.92 cm (ECC 64), where the ECC 64 genotype was significantly higher than the other genotypes (*p* < 0.05). The ECC 64, ECC 90, and ECC 25 genotypes had higher values of CTo, with 27.92 cm, 26.94 cm, and 26.86 cm, respectively (Table 4).

In the early tapping phase, the ECC 25, ECC 64, and ECC 83 genotypes had higher means of dry rubber yield (DRY), with 5.57, 3.76, and 3.30 g.tree^−1^.tap^−1^, respectively. These values were higher than that of IAN 873 (*p* < 0.05). On the other hand, the ECC 29, ECC 35, and ECC 66 genotypes showed lower DRY means, with values of 0.74, 0.42, and 1.09 g.tree^−1^.tap^−1^, respectively, although these values did not differ significantly from that of IAN 873, which had a mean of 1.88 g.tree^−1^.tap^−1^ (Table 4).

The dry rubber content (DRC) was significantly higher in the IAN 873, ECC 25, ECC 73, ECC 29, ECC 64, and ECC 66 genotypes, with means of 32.83, 32.45, 32.42, 31.44, 31.38, and 20.75%, respectively. The ECC 35 and ECC 90 genotypes had lower DRC values (Table 4).

### 2.4. Resistance to SALB

The severity of a *P. ulei* attack on *H. brasiliesis* is evident in Figure 2. Figure 2a,b represent minor and moderate attacks, respectively, which are observed in genotypes with low susceptibility to SALB. Figure 2c,d represent severe and very severe attacks, respectively. Severe attacks occurred in clones with high susceptibility to SALB, as was the case with IAN 873.

According to Figure 3, there was a low SALB incidence in the first year, but in years two and three, there was a significant increase in susceptibility to *P. ulei*. In young leaves, mean scores of AT1 and TR were lower than 1. Some genotypes had mean scores > 1 and >3 for AT2 and ST, respectively. In terms of susceptibility, it was evident that the ECC 35 genotype had the highest SALB mean scores, whereas ECC 73 had the lowest scores.

For AT1, the ECC 64, ECC 29, and ECC 90 genotypes had lower attack intensity, with AT1 < 1. Despite the significant increase in the attacks during the third year, the scores were lower than 1 (Figure 3a). The TR means were statistically different between genotypes, with a significant increase in the second and third years. The ECC 64, ECC 90, and ECC 29 genotypes were the least susceptible, with TR = 0. On the other hand, the ECC 60, ECC 83, and ECC 35 genotypes showed higher sporulation intensities, especially in the third year. However, TR means < 1 were observed in all genotypes over the years (Figure 3b).

In mature leaves, a significant increase in the AT2 means were observed between the genotypes and between the years. The ECC 35 genotype was the most susceptible, with an average AT2 score > 1, followed by IAN 873, ECC 83, ECC 60, ECC 25, and ECC 66. However, the highest score was observed in the third year, where ECC 35 had the highest severity scores (AT2 > 2) and was significantly superior to other genotypes in all years. Furthermore, for the second year, ECC 35 exhibited the highest susceptibility to SALB. On the other hand, the ECC 64, ECC 73, ECC 29, and ECC 90 genotypes were the least susceptible, with an average AT2 < 1 (Figure 3c).

With respect to the sexual phase of *P. ulei*, it was found that the ECC 35 and IAN 873 genotypes had higher formation of stroma (ST > 2). In the third year, ECC 35 was the most susceptible, with mean ST scores > 3, followed by IAN 873, ECC 60, ECC 25, ECC 66, ECC 83, and ECC 60 (all, ST > 2). In the second year, ECC 25, ECC 83, and ECC 60 had mean ST scores > 2. Overall, ECC 64 was the least susceptible, with an average ST < 1. The ECC 73, ECC 64, and ECC 90 genotypes had mean ST scores <1 and were the least susceptible in the third year (Figure 3d).

The co-inertia analysis between the data matrices did not show significant covariation (*p* > 0.05). This indicates that the co-variation structure between early performance and SALB resistance was not entirely similar to the structures obtained in the individual analyses of each variable matrix. However, the first two axes of the co-inertia analysis explained 88.36% and 11.1% of the total variability, respectively (Figure 4a,b, respectively), indicating that the simultaneous analysis of both matrices was suitable for explaining the observed ordering pattern in the 10 genotypes of *H. brasiliensis.*

The most susceptible genotypes positioned themselves in the lower-left quadrant of Figure 4c, characterized by lower vigor and lower yield (DRY and DRC, respectively) (Figure 4d), as well as higher scores of AT2 and ST. Additionally, a positive association was observed between the ECC 60 genotype and the AT1 and TR variables (i.e., a higher susceptibility in young leaves) (Figure 4e). On the other hand, the ECC 64 genotype showed higher values of CTo and CTpty3 (Figure 4d). This genotype had greater vigor and lower SALB susceptibility, positioning itself in the lower-right quadrant of Figure 4c. The ECC 25, ECC 29, ECC 73, ECC 64, and ECC 90 genotypes were located towards the right side of the plot and had the best results in terms of growth and early yield, along with low susceptibility to SALB.

## 3. Discussion

This study evaluated the resistance to SALB, growth, and early yield of nine Colombian elite genotypes of *H. brasiliensis* from the ECC-100 series (Elite, Caquetá Colombia) and IAN 873 (control) under conditions of low SALB pressure.

The climatic characteristics of the study area (i.e., average temperature of 25.1 °C, average relative humidity in the driest months ranging from 69 to 75%, and precipitation of 2514.6 mm) are typical of regions with low SALB pressure [9,23,26,31,38]. These climatic parameters are unfavorable for the production of *P. ulei* spores [39] but they are optimal for rubber tree cultivation [9,23,24,25].

For the three years of the study, between November and March, there was a reduction in precipitation and relative humidity, along with an increase in temperature compared to the period from April to October. According to Sterling et al. [30], these climatic variations are optimal for the growth of *H. brasiliensis* in areas with no escape from SALB, resulting in more vigorous trees in the first few years after planting. On the other hand, Rivano et al. [24] found that climatic variations allowed for the foliage recovery of some clones during the dry season, and, under suitable phytosanitary conditions, they were not affected by SALB in a suboptimal rubber cultivation area in Ecuador. Furthermore, Sterling et al. [31] found that the climatic conditions in San Vicente del Caguán contribute to SALB resistance for introduced rubber tree genotypes from Brazil, Peru, and Guatemala. Our results support the idea that these climatic conditions are favorable, promoting better vigor and greater resistance to SALB for Colombian rubber tree genotypes.

In this study, the circumference of the trunk (CT) was higher than that reported in previous studies in the three first years of growth [30,31]. In escape SALB conditions, Sterling et al. [31] found CTpt values that ranged from 6.50 to 8.61 cm in the first year of planting. In contrast, in non-escape SALB conditions, Sterling et al. [30] reported CT values that ranged from 3.64 to 5.98 cm in the first year for ECC-100 series genotypes. Our results showed CTpt values of between 8.73 and 10.69 cm in the first year, higher than those reported in the Colombian Amazon region.

In this study, in the second year, the average CTpt values ranged from 13.33 to 20.44 cm, similar to those reported by Sterling et al. [40]. For the third year, the average CTpt values ranged from 20.23 cm to 27.48 cm, which were higher than those reported by Gireesh et al. [41] in India, where 20 promising genotypes had a girth of between 15.3 and 23.0 cm. The ECC 64 and ECC 25 genotypes were the most vigorous, with an average annual trunk circumference increase of 7 to 8 cm, similar to the results of Rivano et al. [24], who observed an 8 cm increase in circumference for the two most vigorous clones in a suboptimal rubber cultivation zone in Ecuador. This demonstrates the high potential of ECC-100 series genotypes for zones with low SALB pressure.

The rubber tree immaturity period ranges from four to eight years before conventional tapping is applied. Genotypes with high CT in the early years of planting reach the tapping period faster, allowing for early exploitation and early cost recovery for tree maintenance [41,42,43]. Therefore, one of the goals of rubber tree breeders is to identify genotypes that combine high yield and vigorous growth in early evaluation [44]. In this regard, the Colombian elite genotypes evaluated in this study showed a high CT in the first three years, similar to the highest-performing clones evaluated by Rivano et al. [24] and superior to those evaluated by Gireesh et al. [41].

Climate conditions and clonal variation have a direct impact on the phenotypic expression of *H. brasiliensis* [45,46]. This indicates that genotype performance depends not only on the genetic nature of the trees but also on the environmental variations of the planting site [47]. In this study, a greater growth of ECC-100 series genotypes was observed compared to previous studies conducted in areas with favorable conditions for SALB [30]. These findings highlight the importance of considering both the selection of suitable genotypes and environmental conditions to maximize the performance of *H. brasiliensis*.

The dry rubber content (DRC) is an important parameter for assessing latex quality, as it is one of the main components that directly or indirectly contribute to rubber yield, and it determines the amount of rubber in a latex sample [48,49]. In this study, genotypes had a DRC that ranged from 26.24% to 32.83%. These results are lower than those reported by Mendoza-Vargas and Jiménez-Forero [50] (i.e., DRC of 34.3%) in a promising clonal collection from the FX series in Cundinamarca, Colombia. Furthermore, the percentages obtained are lower than those reported by Ajith et al. [51] in two locations in India (i.e., DRC ranged from 35.23% to 48.9%). The DRC values of IAN 873 (control) were also lower than those reported by Quesada-Méndez et al. [48] in genotypes three years after planting (i.e., DRC of 40.91%) and lower than those reported by Mendoza-Vargas and Jiménez-Forero [50] (i.e., DRC of 41.3%).

The differences observed in DRC between the genotypes were mainly due to variations in the colloidal composition of latex, which depends on soil conditions, climate, tapping duration, diseases, and plant material. These factors primarily affect the DRC composition. Overall, the DRC values ranged between 27 and 45% [52,53]. This indicates that the results of our study fall within optimal composition ranges.

Higher DRC values represent greater productivity in terms of dry rubber yield. However, elevated percentages also imply an increase in the viscosity of the colloidal latex composition, which can lead to blockages in the laticifer mantle and obstruction of the latex flow from the tapping panel [48,54], which can reduce dry rubber yield (DRY). In this study, DRY values ranged from 0.42 to 5.57 g.tree^−1^.tap^−1^, which are lower than those reported by Ajith et al. [51] (i.e., DRY of between 2.88 and 54.5 g.tree^−1^.tap^−1^).

Assessing the temporal dynamics of SALB resistance is crucial to determining optimal areas for crop establishment [30]. This study revealed different susceptibility patterns depending on the year and genotype. The severity of the attacks on both young and mature leaves increased significantly over the years, especially in the third year and in the ECC 35, ECC 60, and IAN 873 genotypes. These results are consistent with those of Sterling et al. [30], who found that the ECC 35 and ECC 60 genotypes were affected the most in non-escape SALB areas. Furthermore, the ECC 64, ECC 73, ECC 90, ECC 25, and ECC 29 genotypes showed lower susceptibility, positively correlating with higher vigor and early yield. These results differ from those of Sterling et al. [30], who found that the genotypes with higher resistance did not necessarily exhibit greater vigor.

With respect to asexual or conidial sporulation, it occurs on young leaves, and during this phase, conidia can disperse for short distances during dry periods [55], but it does not ensure pathogen survival [56]. However, in very rainy seasons, conidia can be particularly abundant [55]. The San Vicente del Caguán site in this study is characterized by low precipitation and well-defined dry periods (i.e., semi-humid warm climate), which is unfavorable for the production of *P. ulei* conidia (i.e., low TR scores).

In relation to sexual sporulation, which provides quantitative information about the intensity of ascospores on mature leaves [36], the attack severity increased in the second and third years. In the second year, a moderate level of attack was evident for all genotypes, with the exception of the ECC 29 and ECC 64 genotypes, which had lower attack levels. In the third year, there was an increase in susceptibility; however, moderate attack levels were still recorded for all genotypes, with the exception of the ECC 35 genotype, which suffered severe attacks. Overall, we observed an increase in ascospore production over the years. However, the ECC 64, ECC 73, ECC 90, ECC 25, and ECC 29 genotypes were the least susceptible. The SALB scores in the first year were similar to those reported by Rivano et al. [36] in the CDC 312, FDR 4575, FDR 5597, and MDF 180 clones, which were classified as completely resistant, and in the third year, the SALB scores were similar to those reported by Sterling et al. [34].

Considering that clones with high yield and SALB resistance are rare [57], this dual-choice criterion is very important in rubber tree breeding programs. Based on these results, the ECC 64, ECC 73, ECC 90, ECC 25, and ECC 29 genotypes are considered potential candidates for establishment in areas with low SALB pressure in the Colombian Amazon owing to their low susceptibility to SALB and high performance in the first three years of growth.

## 4. Materials and Methods

### 4.1. Study Area

#### 4.1.1. Location

The study was conducted from July 2017 to June 2020 on the Parcela No. 12 farm, located in the Buenos Aires rural settlement area in the Municipality of San Vicente del Caguán, Caquetá, Colombia. This site is located in the southern part of the Colombian Amazon (02°01′42.62″ N and 74°54′38.95″ W, at an elevation of 346 m above sea level. According to the classification by IGAC [58], the landscape in the study zona is hilly, characterized by hills with heights of less than 300 m and slopes ranging from 7 to 12 percent. The region is a zone with low SALB pressure, making it a promising region for rubber cultivation [26,33].

#### 4.1.2. Soils

The soils in Caquetá have unfavorable characteristics such as high compaction, low fertility and infiltration, poor drainage, shallow depth, and a pH range of between 4.5 and 5.8. Furthermore, these soils have high levels of aluminum saturation and low contents of sodium (Na), phosphorus (P), potassium (K), and magnesium (Mg) [58].

The study area had a very acidic pH of 5.05, an organic matter (OM) content of 1.21%, and an organic carbon (OC) content of 0.715%. Saturation levels were 17.51% (Ca), 1.18% (K), 4.99% (Mg), and 2.23% (Na). Elemental contents were as follows: iron (Fe) (161.59 mg kg^−1^), manganese (Mn) (4.27 mg kg^−1^), copper (Cu) (1.71 mg kg^−1^), zinc (Zn) (0.54 mg kg^−1^), boron (B) (0.19 mg kg^−1^), phosphorus (P) (0.84 mg kg^−1^), sulfur (S) (6.91 mg kg^−1^), and total nitrogen (N) (0.06%).

#### 4.1.3. Climate

The climatic conditions in the study area correspond to a semi-humid warm climate, typical of the tropical wet region. It records an average temperature of 25.4 °C, a relative humidity of 82.3%, and an annual precipitation of 2503 mm [58]. The climate follows a monomodal regime with a transition to a dry period from July to October, followed by a dry period with less rain between November and February and, finally, a rainy period from March to June [59].

Microclimatic data for the study area were collected with a Portable Microclimate Station (Decagon Devices Inc., Pullman, WA, USA), from which monthly averages were calculated between July 2017 and June 2020 (Figure 1).

### 4.2. Experimental Design and Plot Management

A large-scale clonal trial (LSCT) [60] was established using a randomized complete block design (RCBD) with four replications (Fisher blocks). Each treatment consisted of planting 60 trees in single rows, with a planting distance of 7.0 × 3.0 m, equivalent to a planting density of 476 trees per hectare, and the total area of the LSCT was 5.04 hectares. The measured plot unit was 1260 square meters, containing 60 trees organized in 3 rows of 20 trees [30,31,34].

With respect to the management of the experimental plots, the fertilization plan employed by Sterling et al. [61] was applied every six months. Thus, a compound fertilizer (N (15%), P2O5 (15%), K_2_O (15%), CaO (2.2%), and SSO_4_ (1.7%)) was used at a rate of 150 g per plant. Minor elements included N (8%), P_2_O_5_ (5%), CaO (18%), MgO (6%), S (1.6%), B (1%), Cu (0.14%), Mo (0.005%), and Zn (2.5%), applied at a rate of 75 g per plant. Additionally, organic compost was applied at a dosage of 1000 g per plant. Weed control was carried out every three months, and no phytosanitary controls were performed [30,31,34].

### 4.3. Plant Material

Nine Colombian elite *H. brasiliensis* genotypes from the ECC-100 series (Elite Caquetá Colombia) were evaluated: ECC 25, ECC 29, ECC 35, ECC 60, ECC 64, ECC 66, ECC 73, ECC 83, and ECC 90. These genotypes were obtained from asexual propagation (cloning) of plus trees of *H. brasiliensis* from rubber-producing farms in Caquetá, Colombia, through natural cross-pollination and were characterized using molecular markers [30].

Evaluations carried out in small-scale [62] and large-scale clonal trials [30] in non-SALB escape zones demonstrated favorable performance in terms of growth and partial resistance to SALB compared to traditional cultivars introduced in the Colombian Amazon.

In this study, the cultivar IAN 873 was used as a control, which is a clone widely cultivated in Colombia due to its good agronomic performance [63], which contrasts with its high susceptibility to *P. ulei*, as evidenced in the last 15 years mainly in non-SALB escape zones [33].

### 4.4. Evaluations and Data Analysis

#### 4.4.1. Growth and Early Yield

The annual trunk circumference at 1.2 m above the ground was recorded during the pre-tapping phase (CTpt) of all trees from July 2017 to June 2020 and in the early tapping phase in the third year (CTpty3). Then, the CT mean was calculated only for the trees that reached the threshold for extraction (CT ≥ 25 cm) at 1.2 m above the ground at the opening of the tapping panel (circumference of the trunk at opening, CTo) of the trees suitable for early tapping in the third year after planting (modified from Silva et al. [64]).

The dry rubber yield (DRY) (g.tree^−1^.tap^−1^) was recorded during the first six months of tapping, following the methods described by Meenakumari et al. [37] and Silva et al. [64]. The dry rubber content (DRC) as a percentage was measured according to the method described by Lacote et al. [65] and Mendoza-Vargas and Jiménez-Forero [50].

#### 4.4.2. Resistance to SALB

The resistance of *H. brasiliensis* to SALB was assessed monthly in 50% of the trees in each plot (30 trees) between July 2017 and June 2020 using a method adapted from Rivano et al. [36]. The data were analyzed by year. To evaluate resistance to *P. ulei*, both young and mature leaves (stages C and D, respectively) were examined [31].

The severity of the attacks on young leaves (AT1) and mature leaves (AT2) was assessed using a scale adapted by Rivano et al. [36] based on the scale proposed by Chee and Holliday [18]. This scale defines five categories based on the percentage of leaf surface with symptoms or signs of the disease: 0 represents no attack (<1%), 1 represents a minor attack (1–5%), 2 represents a moderate attack (6–15%), 3 represents a severe attack (16–30%), and 4 represents a very severe attack (>30%).

In young leaves, the intensity of conidial sporulation (asexual phase of the fungus) on the lesions formed on the leaf surface at stage C was visually assessed using the variable “type of reaction” (TR). This variable is based on a scale from 1 to 6 proposed by Junqueira et al. [66] and adapted by Mattos et al. [67]. The scores on this scale are as follows: (1) necrotic lesions without spores; (2) non-necrotic lesions without spores; (3) very scarce and heterogeneous sporulation on the underside of the leaf; (4) high, heterogeneous, or partial sporulation on the underside of the leaf; (5) very high and uniform sporulation covering the entire lesion on the underside of the leaf; and (6) very high sporulation covering the entire lesion on both the underside and upper side of the leaf.

The intensity of sexual sporulation of the fungus on the upper side of the leaves at stage D was assessed using the variable “stroma density” (ST). A scale proposed by Rivano et al. [36] was used, defining four scores based on the number of stromata per leaflet: (0) absence of stromata, (1) fewer than 5 stromata, (2) between 5 and 10 stromata, (3) between 11 and 30 stromata, and (4) more than 30 stromata. For all of these variables, the most severely affected leaves on each plant were evaluated and the average of each variable was calculated for each replicate of each genotype [34].

#### 4.4.3. Statistical Analysis

A linear mixed-effects model (LME) was used to analyze the fixed factors genotype, year, and their interaction for the variables CTpt, AT1, AT2, TR, and ST. QQ-plots of the model residuals were used to assess the normality assumption and fitted plot (residual vs. predicted) for the homogeneity of variances. A transformation was applied to the AT1, AT2, TR, and ST variables to normalize the data, using the formula (value + 0.5)^1/2^. The effect of genotype on the growth variables was evaluated at the end of the evaluation cycle. Block and plot effects associated with genotypes within blocks were considered random effects. Residual variance was modeled to adjust the heteroscedasticity by genotype, and residual correlation for successive observations was considered using models for longitudinal data: compound symmetry with heterogeneous variances (CSH), first-order autoregressive with heterogeneous variances (ARH1), continuous first-order autoregressive with heterogeneous variances (CARH1), and unstructured with heterogeneous variances (UNH) [35]. The Akaike (AIC), Bayesian (BIC), and log-likelihood (Log Lik) criteria were used to select the structure of variances and residual correlations [34,35].

To analyze the effect of genotype on variables related to early yield (CTo, DRY, and DRC), a linear mixed-effects model (LME) with independent errors was used. The analyses were performed with the “lme” function (linear mixed-effects models) in the “nlme” package (linear and nonlinear mixed-effects models) [68] in the R language software, version 3.5.1 [69], and the interface in InfoStat v. 2018 [70]. Differences between means of the variables were analyzed using Fisher’s LSD test (α = 0.05).

To explore the relationships between performance (growth and early yield) and SALB resistance in the 10 genotypes, two matrices were constructed with each group of variables, and a Principal Component Analysis (PCA) was performed as an ordination method. The results of the ordination of the two groups of variables were carried out with a co-inertia analysis with the Monte Carlo test [71]. The software used for this analysis was the ADE-4 package [72] in R 3.5.1 [69].

## 5. Conclusions

The results of this study support the idea that, under favorable climatic conditions (i.e., zones with low SALB pressure), the genotypes from the ECC-100 series had higher yields and lower SALB susceptibility, and, therefore, could be successfully established in regions such as the Colombian Amazon. This research contributes to the development of more effective management strategies based on genetic resistance to the disease.

Most tested genotypes showed low susceptibility to *P. ulei*. Over time, the impact of the disease increased significantly; however, when compared to other studies, the susceptibility level was very low (partial resistance to SALB, with mean scores of TR < 2 and ST < 3). Furthermore, the genotypes with lower susceptibility to SALB expressed higher early performance indices (CTpt > 23 cm, DRY > 3 g.tree^−1^.tap^− 1^, and DRC > 31% in the third year after planting). The optimal early responses in yield, growth, and SALB resistance reported in this study, although preliminary, are of interest for the selection of new desirable *H. brasiliensis* genotypes for the Amazon region. However, it is necessary to analyze the productive performance and SALB resistance for a longer period comprising the crop maturity stage before generating a commercial recommendation under the Colombian Amazon region conditions.

## Figures and Tables

**Figure 1 plants-12-03627-f001:**
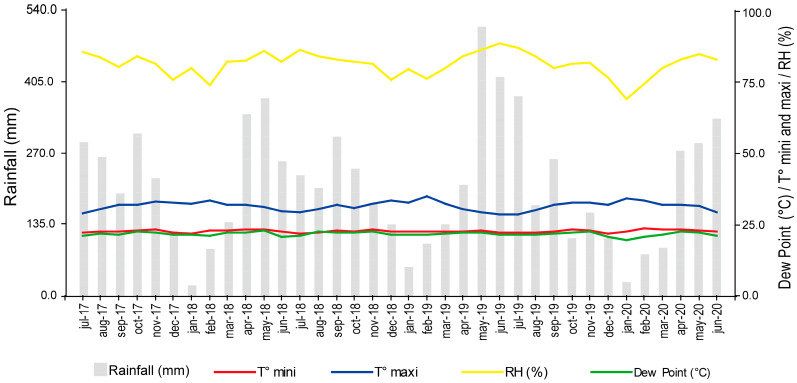
Monthly precipitation, T-max and T-min (°C): maximum and minimum temperature; RH (%): relative humidity (San Vicente del Caguán, Caquetá, Colombia) between July 2017 and June 2020.

**Figure 2 plants-12-03627-f002:**
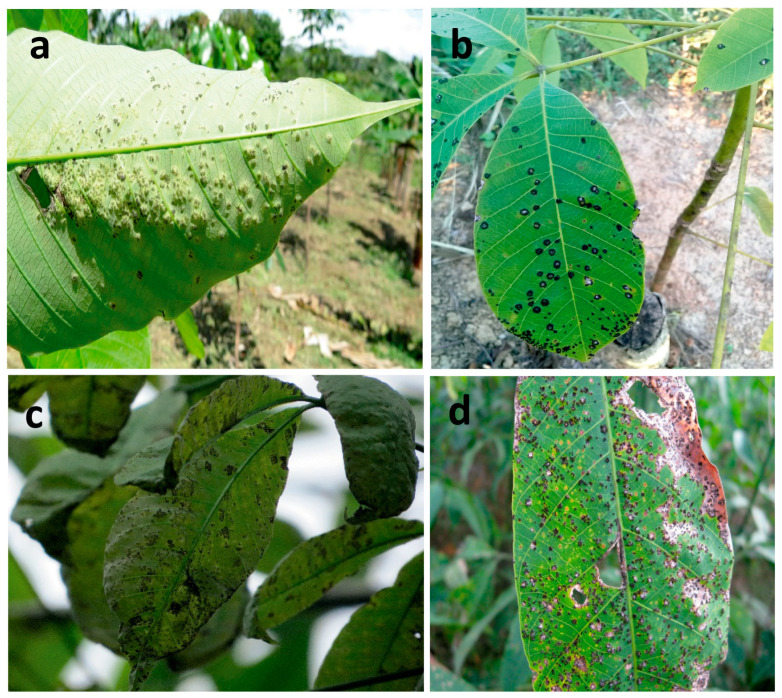
Representation of the attack of *P. ulei* on *H. brasiliensis*. (**a**) AT1 represents a minor attack (1–5%). (**b**) AT2 represents a moderate attack (6–15%). (**c**) AT3 represents a severe attack (16–30%). (**d**) AT4 represents a very severe attack (>30%).

**Figure 3 plants-12-03627-f003:**
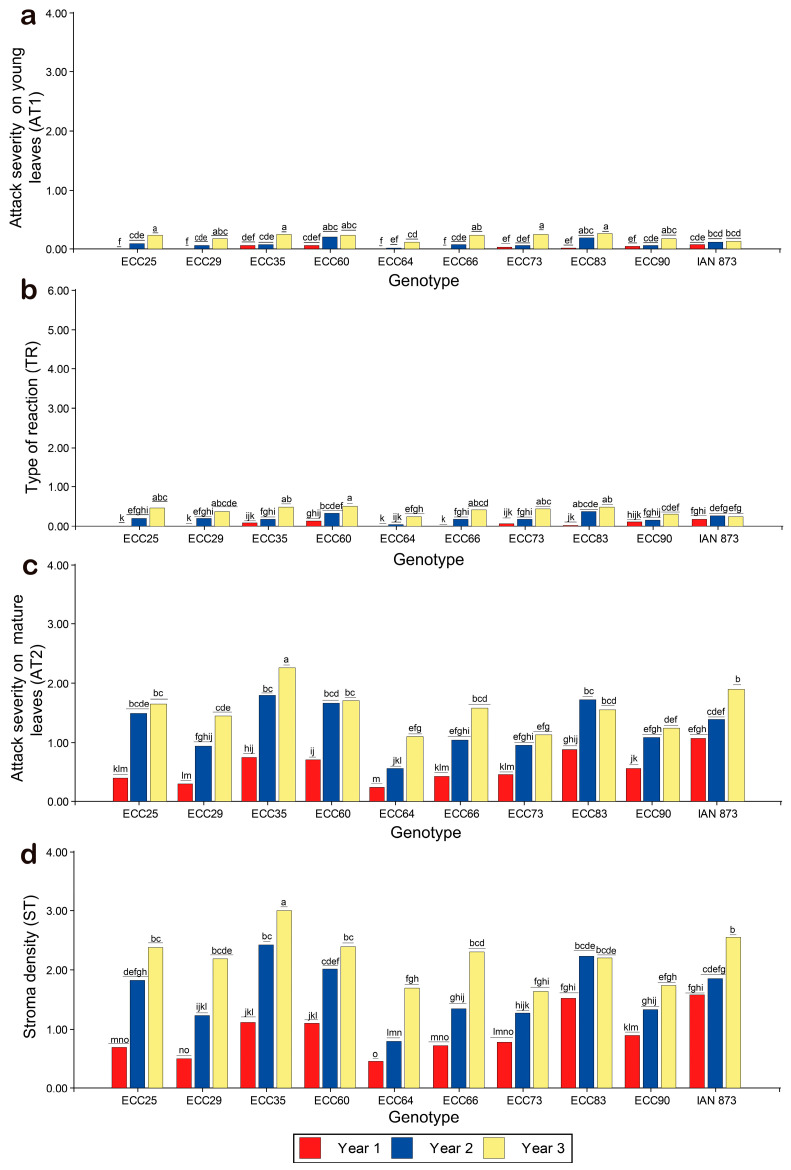
Results of susceptibility to SALB over three years after planting. The following parameters were evaluated: (**a**) attacks of *P. ulei* on young leaves at stage C (AT1); (**b**) type of reaction (TR) to *P. ulei* on young leaves at stage C; (**c**) attacks of *P. ulei* on mature leaves at stage D (AT2); (**d**) stromal density on mature leaves at stage D (ST). Means between the genotypes followed by the same letter in the same year do not differ statistically (Fisher’s LSD test, *p* < 0.05).

**Figure 4 plants-12-03627-f004:**
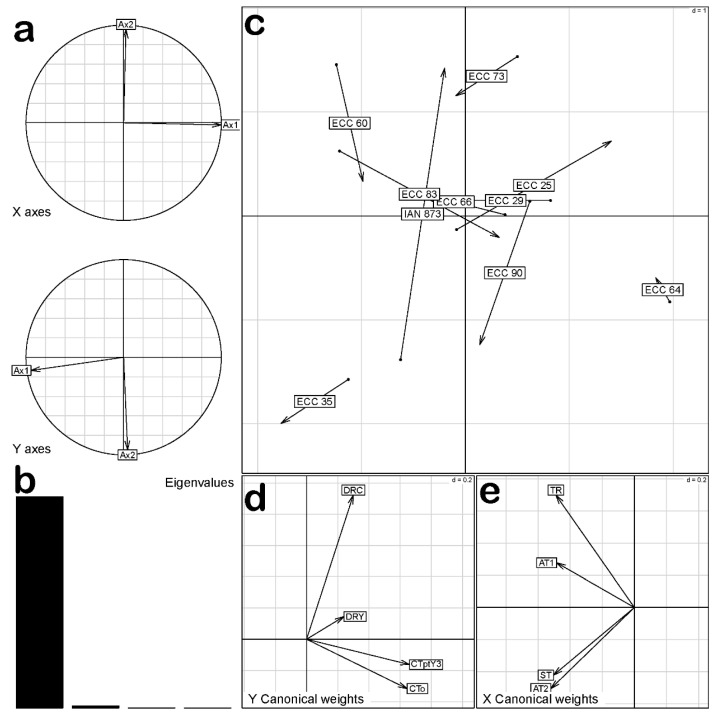
Co-inertia analysis between SALB resistance of 10 genotypes of *Hevea brasiliensis*: (**a**) PCA (Principal Component Analysis); (**b**) captured inertia (%); (**c**) genotype ordination plot; (**d**) growth and early yield projection; (**e**) SALB resistance projection.

**Table 1 plants-12-03627-t001:** Selection criteria for the longitudinal data analysis model for assessing the growth and resistance to SALB.

Model	Compound Symmetric Model with Heterogeneous Variances (CSH)			First-Order Autoregressive Model with Heterogeneous Variances (ARH1)			First-Order Continuous Autoregressive Model with Heterogeneous Variances (CARH1)			Unstructured Model with Heterogeneous Variances (UNH)		
Variable	Log Lik	AIC	BIC	Log Lik	AIC	BIC	Log Lik	AIC	BIC	Log Lik	AIC	BIC
CTpt	−175.38	420.77	508.26	−170.39	410.77	498.27	−170.39	410.77	498.27	−165.18	404.35	496.84
AT1	159.78	−235.6	−130.6	159.97	−236	−131	159.97	−235.95	−131	161.4	−234.8	−124.81
AT2	63.18	−56.36	31.13	63.29	−56.59	30.91	63.29	−56.59	30.91	64.22	−54.44	38.06
ST	48.8	−27.59	59.9	48.68	−27.37	60.13	48.68	−27.36	60.13	49.65	−25.3	67.2
TR	97.9	−125.8	−38.31	97.1	−124.2	−36.7	97.1	−124.19	−36.7	101.28	−128.57	−36.07

Log Lik, log-likelihood; AIC, Akaike criterion; BIC, Bayesian criterion [35]; CTpt, circumference of the trunk in the pre-tapping phase; AT1, attack severity on young leaves; AT2, attack severity on mature leaves; TR, type of reaction on young leaves (conidial sporulation intensity, TR); ST, stroma density on mature leaves [31,34,36].

**Table 2 plants-12-03627-t002:** Analysis of variance (ANOVA) in growth, SALB resistance, and early yield for effects of genotype, year, and genotype × year interaction.

Variable	DF1	DF2	DF3	Genotype	Year	Genotype × Year
CTpt	9	2	18	4.42 (0.0001)	714.72 (<0.0001)	2.23 (0.0074)
AT1	9	2	18	5.35 (<0.0001)	89.39 (<0.0001)	2.98 (0.0004)
AT2	9	2	18	16.19 (<0.0001)	256.90 (<0.0001)	2.68 (0.0012)
ST	9	2	18	19.30 (<0.0001)	215.16 (<0.0001)	3.08 (0.0002)
TR	9	2	18	3.50 (0.0010)	116.21 (<0.0001)	2.09 (0.0125)
DRC	9	-	-	10.14 (<0.0001)	-	-
DRY	9	-	-	18.85 (<0.0001)	-	-
CTo	9	-	-	5.69 (<0.0001)	-	-

CTpt, circumference of the trunk in the pre-tapping phase; AT1, attack severity on young leaves; AT2, attack severity on mature leaves; TR, type of reaction on young leaves (conidial sporulation intensity, TR); ST, stroma density on mature leaves [31,34,36]. DRC, dry rubber content; DRY, dry rubber yield; CTo, circumference of the trunk at opening [31,37]; DF1, degrees of freedom for the factor genotype; DF2, degrees of freedom for the factor year; DF3, degrees of freedom for interaction genotype × year; -, not applicable.

**Table 3 plants-12-03627-t003:** Annual circumference of the trunk (cm) in the pre-tapping phase (CTpt) for 10 genotypes of *Hevea brasiliensis*.

Genotype	Year 1	Year 2	Year 3
ECC 25	9.37 ± 0.49 ^a,b^ k	18.33 ± 0.88 def	25.75 ± 1.24 ab
ECC 29	9.43 ± 0.49 k	16.98 ± 0.88 fgh	22.80 ± 1.24 bc
ECC 35	7.78 ± 0.49 l	13.33 ± 0.88 i	20.23 ± 1.24 cde
ECC 60	9.03 ± 0.49 k	14.89 ± 0.88 hi	21.44 ± 1.24 c
ECC 64	10.69 ± 0.49 j	20.44 ± 0.88 cd	27.48 ± 1. 24 a
ECC 66	8.73 ± 0.49 kl	15.31 ± 0.88 ghi	22.54 ± 1.24 bc
ECC 73	9.44 ± 0.49 k	16.66 ± 0.88 fgh	22.84 ± 1.24 bc
ECC 83	9.62 ± 0.49 jk	17.31 ± 0.88 efg	23.61 ± 1.24 bc
ECC 90	9.02 ± 0.49 k	15.83 ± 0.88 gh	23.63 ± 1.24 bc
IAN 873	9.82 ± 0.49 jk	16.28 ± 0.88 fgh	22.41 ± 1.24 bc

^a^ Standard error; ^b^ Means in each column with the same letter do not differ statistically (Fisher’s LSD test, *p* < 0.05).

**Table 4 plants-12-03627-t004:** Circumference of the trunk (cm) at opening (time of commencement of tapping or 3rd year) (CTo); DRY, dry rubber yield; DRC, dry rubber content in the early tapping phase for 10 genotypes of *Hevea brasiliensis*.

Genotype	CTo (cm)	DRY (g.tree^−1^.tap^−1^)	DRC (%)
ECC 25	26.86 ± 0.35 ^a,b^ b	5.57 ± 0.43 a	32.45 ± 0.44 a
ECC 29	26.49 ± 0.37 b	0.74 ± 0.18 c	31.44 ± 2.29 a
ECC 35	25.00 ± 2.45 b	0.42 ± 1.14 c	26.24 ± 0.76 b
ECC 60	25.03 ± 0.93 b	1.21 ± 0.43 c	30.43 ± 0.70 ab
ECC 64	27.92 ± 0.26 a	3.76 ± 0.66 b	31.38 ± 1.01 a
ECC 66	25.54 ± 0.78 b	1.09 ± 0.43 c	30.75 ± 1.10 a
ECC 73	25.62 ± 0.56 b	2.23 ± 0.43 bc	32.42 ± 1.18 a
ECC 83	26.09 ± 0.45 b	3.30 ± 0.24 b	30.53 ± 1.98 ab
ECC 90	26.94 ± 0.57 b	2.96 ± 0.81 bc	28.83 ± 3.05 ab
IAN 873	25.47 ± 0.60 b	1.88 ± 0.28 c	32.83 ± 0.10 a

^a^ Standard error. ^b^ Means in each column with the same letter do not differ statistically (Fisher’s LSD test, *p* < 0.05).

## Data Availability

Data are available from the authors upon request.

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
