# Peer review of "Performance- and Resistance-Related Early Responses of Colombian Elite Rubber Tree Genotypes under Low Pressure of South American Leaf Blight: Implications for Disease Management in the Amazon"

_plants, 2023, doi:10.3390/plants12203627_

Round 1

Reviewer 1 Report

The manuscript reports on the analysis of responses of 11 lines of Hevea brasiliensis (the source of rubber) against South America Leaf Blight, compared to a current cultivar.  This finds a range of responses of the lines against this disease, as well as other agronomically important properties.  The authors do provide a caveat that the plants need to growth through to maturity before final conclusions could be reached.  The research should appeal to those engaged in breeding and/or analysis of disease resistance in crops.

A few points for consideration are as follows.

(1)    Due to the structure of the paper, the Results section starts without enough background to understand the experiments that were conducted.  This is especially the case for section 2.2.  It would be better to provide a least a few sentences of background since many readers will continue to this section straight from the Introduction.

(2)    Related to point 1, the abbreviations for the traits being measured appear as abbreviations with no context.  It makes the paper hard to understand.  If abbreviations can be avoided, then best to do such.  For example, in figure 2 the y-axes could be in full.

(3)    Is it possible to make these conclusions based on a single field site?  It seems that the responses across a wider geographic region would be a better guide of selection of the next lines.

(4)    In numerous places in the Dicussion comparisons are made between studies, but is this not of limited value given they were performed under different environmental conditions or with different plant material?

There are a few typographical errors to fix.

Line 4: in the text, ‘Leaf Blight’ is in capitals but here in lower case.

Table 3: check style for ECC 25 and ECC 29 to remove commas in ‘0,49’.

Table 4: check style to remove gaps such as between for ECC 25 ‘0. 35’ and ‘0. 44’.

Line 175: delete space ‘ECC 60 had’.

Line 291: spelling ‘Furthermore’.

Line 297: probably ‘pathogen’ is better than ‘disease’.

Line 313: ‘SABL-resistance are rare’.

Line 317: ‘having’ for ‘have’.

Lines 364-365: check subscript on the chemicals used.

Line 460: delete ‘The’ so it is ‘Most genotypes showed…’.

References: check formatting like italics on the species names.

Reviewer 2 Report

The Authors report the results of a field test to evaluate the agronomic performances and susceptibility to Pseudocercospora ulei infections of slected clones of rubber tree (Hevea brasiliensis) in the Amazonian region of Colombia.

The experimental design is simple but appropriate. Statistical analysis of experimental results is adequate. results are of relevant practical interest.  Ovrall the article is well conceived and written.

I suggested some text editings and a minor change in the title ( for details see notes in the attached PDF). 

The English style is adequate and meets international standards.

Reviewer 3 Report

The manuscript titled “Performance-and-resistance-related early responses of Colombian elite rubber tree genotypes under low pressure of South American leaf blight: implications for disease management in the Amazon" by Galindo-Rodríguez et al. contributes to the selection of new Pseudocercospora ulei-resistant Colombian genotypes of H. brasiliensis, an important producer of natural rubber, by evaluating the growth, early yield, and resistance to Pseudocercospora ulei of nine Colombian elite genotypes from the ECC-100 series and a control (IAN 22 873) in a large-scale clone trial in the Colombian Amazon during the period 2017 - 2020.

Introduction:

- please elaborate on the nature of escape zones. What makes them different from high pressure zones?

- line 66: please define FAO

- please describe the fungal agent, Pseudocercospora ulei, in more detail

- overall, the introduction is written well and contains most of the relevant information

Methodology:

- line 364-: please use the correct abbreviations for the chemical formulae (i.e., subscripts like P2O5)

- the reviewer appreciates the detail in which the authors describe their methodology

Results:

- line 111-: please define all abbreviations after their first use

- providing some representative photographs of the fungus infecting different H. brasiliensis clones (and different attack strengths) would be helpful to better grasp the otherwise abstract data presented by the authors.

Discussion:

- very well written and referenced, no criticisms

Recommendation:

a solid contribution that merits reconsideration for publication in Plants after a Minor Revision.
